# Implementation of Universal Design for Learning in Low- and Middle-Income Countries: 'I Thought These Principles Could Have Been Written by Me'

**Judith McKenzie** [1,*], **Amani Karisa** [2] **and Callista Kahonde** [3]

1 Including Disability in Education in Africa (IDEA) Research Unit, Department of Health and Rehabilitation Sciences, Faculty of Health Sciences, University of Cape Town, Cape Town 7701, South Africa
2 Human Development Theme, African Population and Health Research Center, Nairobi 00100, Kenya; amani.karisa@uct.ac.za or mkarisa@aphrc.org
3 Department of Global Health, Centre for Disability and Rehabilitation Studies, Stellenbosch University, Stellenbosch 7600, South Africa; ckkahonde@sun.ac.za
* Correspondence: judith.mckenzie@uct.ac.za

**Abstract:** UNESCO's Global Education Monitoring Report (2020) strongly recommends the adoption of Universal Design for Learning (UDL) at the government level so that it becomes an integral part of countries' inclusive education policies. However, UDL has largely been developed in high-income countries with technology as a central theme. The question is whether the purported benefits of the UDL approach can translate into low and middle-income country (LMIC) contexts. This study explores the relevance and fit of UDL to LMIC contexts by use of interviews (either individual or group) with 12 representatives of international agencies, non-governmental organizations (NGOs), and UDL experts who have experience in providing inclusive education services in LMICs. Three themes are reported: understanding of UDL, UDL supporting inclusive education, and UDL in teacher-training initiatives and capacity building. The concept of UDL is not new in LMICs, although the name may be. The potential for UDL to support inclusive education in LMICs is recognized. UDL needs to be mainstreamed in teacher training programs, and its implementation should be adapted to respond to the contextual realities of LMICs. The encounter between UDL proponents from high-income countries and education stakeholders in LMICs should be mutually enriching rather than imposing the approach from high-income countries on those in LMICs.

**Keywords:** disability; inclusion; LMIC; UDL





## 1. Introduction

Since at least 1994, when the Salamanca Statement [1] advocated for the adoption of an inclusive education system to cater to the needs of all children, including those with disabilities, there has been a progressive broadening and deepening of the concept of inclusion. In terms of broadening, the scope of inclusive education has grown to incorporate a wide range of marginalized groups, not just those with disabilities. This is best illustrated in Sustainable Development Goal (SDG) 4, which aims to: "ensure inclusive and equitable quality education and promote lifelong learning opportunities for all" [2]. In terms of deepening, the policy, processes, and strategies for inclusion have been debated, and extensive literature exists on barriers and facilitators of inclusive education. One strategy that is increasingly gaining momentum is the approach of universal design for learning (UDL), which is promoted to operationalize inclusive education by enabling flexible classroom strategies that meet the needs of a wide range of diverse learners.

UDL is an instructional approach that addresses barriers to learning by creating educational environments that cater to a diverse range of learning needs. UDL recognizes that everyone learns differently and that there is no such thing as the average learner [3].

UNESCO's Global Education Monitoring Report strongly suggests that UDL should be embraced at the governmental level, integrating it into the inclusive education policies of nations [4]. However, UDL has largely been developed in high-income countries with technology as a central theme. The question is whether the purported benefits of the UDL approach can translate into low and middle-income country (LMIC) contexts. This paper sought to draw on the perspectives of key informants on the applicability and usefulness of the UDL approach in facilitating access, participation, and achievement of all students in an inclusive education framework in LMIC. In this paper, we interrogate the applicability of UDL to LMIC contexts through the analysis of these interviews. We begin with a brief overview of UDL and consider the literature available from LMIC on this topic. We then present our study methodology and findings before concluding with some recommendations for the use of UDL in LMIC.

*What Is UDL?*

In its simplest form, UDL is based upon the application of three principles that emerged from a synthesis of research on the neurological basis of learning developed by the Center for Applied Special Technology (CAST) in the USA and their guidelines [5,6]. The three principles are (1) multiple means of representation, which entails providing multiple, flexible methods of presentation to support different ways of knowledge and information acquisition by learners. (2) Multiple means of engagement, which is about offering a variety of flexible engagement choices that enable learners to enhance their involvement and curiosity about the world through a range of accessible learning activities suited to diverse abilities. (3) Multiple means of action and expression whereby learners are offered a variety of adaptable approaches to action and expression by diversifying the methods through which they can demonstrate their knowledge [7]. The principles, which were originally developed to support learners with disabilities, are now acknowledged as crucial for providing an equitable learning opportunity to all students in a classroom [4].

As inclusive education is frequently viewed in narrow terms as an approach only for the inclusion of learners with disabilities, the same happens with UDL [8]. However, UDL is a practice aimed at the development of all learners, irrespective of whether they face barriers to learning. It uses the three principles to make learning and teaching accessible to the broadest possible range of diversity rather than catering to the non-existent "average learner" [9]. UDL recognizes that everyone learns differently, and it has been promoted as an instructional strategy that can address systemic inequality and discrimination, which may arise from the intersectionality of a diverse range of disadvantages (e.g., racial inequality, gender, socio-economic background, disability) [3].

A recent review of the existing literature on UDL in LMIC [8] revealed several important key themes that formed the basis for the development of the interview guide in this study. These themes included the need for capacity building not only for teachers but also for education officials who might be tasked with implementation: concerns around the levels of technology associated with the implementation of UDL; and the extent to which UDL is perceived to be a response to disability exclusively rather than a holistic approach for all learners. There was also evidence from the review that communities and families are not seen as part of implementation and that the potential for UDL to address systemic inequality remains narrowly applied to disability issues. Indeed, one of the founders of UDL, David Rose [10], has expressed the view that UDL has not gone far enough in addressing inequality and discrimination and that the approach needs to be reviewed to address issues of identity, inequality, and social justice. He critiqued the UDL guidelines as offered by the original three principles, based on the fact that they focus more on the barriers to the learners' ability to learn and barely pay attention to the systemic and institutional "identity barriers" that are based on gender, race, class, language, ethnicity, disability, and others. Together with his colleagues at CAST, they advocate for redress in what they call "cracks in the foundation of UDL" and a shift from UDL guidelines' emphasis on weaknesses rather than strengths of the learners [11]. In their view, a revised version of UDL must

recognize the strengths of the learner and the community from which the learner comes, which shapes their identity.

The philosophical underpinnings of UDL, therefore, continue to evolve and to recognize the important role of the learner's identity and the broader context. It is in this spirit that we undertook the current study, examining the applicability of UDL to the LMIC context. This becomes an urgent matter as UNESCO's Global Education Monitoring Report 2020 strongly recommends the adoption of UDL at the government level so that it becomes an integral part of countries' inclusive education policies [4]. The specific question addressed by the study is: What is the relevance and fit of UDL to LMIC contexts?

## 2. Materials and Methods

Key informants in education development work and UDL in LMIC were identified by the research team and their respective organizations, Christian Blind Mission International (CBM) and University of Cape Town (UCT). These participants were selected purposively based on their job portfolios and publications in relation to education as well as practical field experience, technical knowledge, and expertise in UDL. The following inclusion criteria were applied:

- Recognized expertise and experience in UDL.
- Employed by an organization promoting inclusive education in low- and middle-income countries.
- Experience of a project which supports the implementation of UDL in LMICs.
- Representing different global regions.

Seven interviews (individual or group) were conducted with a total of twelve key informants. For confidentiality purposes, the names of the participants and the organizations to which they belong are not given in this analysis. Codes are used to refer to the participants. Table 1 shows the participant codes, the type of organization to which the participants belong, and the data collection method used.

**Table 1.** Details of participants.

| Participant Code | Organization | Data Collection Method |
|:---:|:---:|:---:|
| P1 | Intergovernmental organization | Individual interview |
| P2 | University in Zimbabwe | Individual interview |
| P3 | | |
| P4 | NGO in India | Group interview |
| P5 | | |
| P6 | | |
| P7 | Independent UDL researcher from USA involved in the training of UDL practitioners in Africa and Asia | Individual interview |
| P8 | NGO in the USA | Group interview |
| P9 | | |
| P10 | International NGO | Individual interview |
| P11 | Intergovernmental organization | Group interview |
| P12 | | |

We adopted a qualitative methodology, seeking different perspectives and understanding of the topic of UDL in low to middle-income countries. This approach was especially relevant as we were concerned with the context in which the implementation of UDL takes place rather than its frequency or effectiveness, which might be explored in further research. The focus was on generating a description of the phenomenon (UDL) in context

(low to middle-income countries) with reference to a small number of cases [12]. The study made use of a phenomenological approach, collecting data in a relatively unstructured way from participants who were considered to be in a position to provide us with rich data on the topic. The phenomenological approach assumes that individuals have unique and subjective experiences of the world. It emphasizes understanding the world from the perspective of the participants, acknowledging that reality is constructed through personal experiences [13]. A semi-structured interview schedule was developed based on the research questions and guided by the findings of the literature review by McKenzie et al. [8] (as discussed in the introduction section). The interview questions were related to the participants' lived experiences and understanding of UDL and how it supports inclusive education. Appendix A shows the interview schedule.

The study was approved by the Human Research Ethics Committee of the Faculty of Health Sciences at the University of Cape Town, South Africa (HREC148/2021). All interviewees were given information about the project and had the opportunity to have any questions answered before signing a letter of informed consent. The interviews were conducted virtually via Zoom by members of the research team, all of whom have experience in qualitative interviewing techniques. All the interviews were conducted in English. We used both individual interviews and group interviews as some of the participants preferred to participate in pairs or as a group, depending on how they worked within their organization. Others who had implemented UDL as consultants or in their own capacity participated as individuals. The individual interviews lasted, on average, 40 min, while the group interviews lasted, on average, 70 min. The interviews were recorded with the permission of the participants. The recordings were then transcribed manually by the researchers, or the transcripts were obtained from Zoom, which were then checked for completeness and corrected as necessary by comparing them with the audio recordings.

Initially, data were analyzed deductively [14] based on the topics covered by the interview questions and guided by findings of a literature review in LMIC by McKenzie et al. [8]. Data analysis entailed reading the participants' responses to the interview questions and coding these manually. The researchers undertook the coding independently of one another to enhance the confirmability of the findings. The codes that had similar traits were then grouped to build themes [14]. The researchers then identified sub-themes within these themes through an inductive process. The content of the themes and subthemes from the different transcripts were then compared and grouped into three themes and their subthemes by the research team working collaboratively.

## 3. Results

The themes and sub-themes are presented in Table 2. The findings are then presented in narrative form, with some direct quotations extracted from the transcripts to illustrate the themes. Each quote is identified by a code that refers to a specific participant.

### 3.1. Understanding of UDL

This theme describes the participants' understanding of UDL, as gleaned from their personal journeys towards implementing UDL, which for many was not a known concept within their work contexts when they began their careers. They all started from the point where they valued the inclusion of students with diverse learning needs, even before they knew about UDL. This made it easier for them to embrace and implement UDL and to continue promoting it in their current roles. They also understood UDL as a tool to facilitate a shift from focusing on perceived learner deficits to focusing on the system from the classroom to the policy level.

**Table 2.** Themes and sub-themes.

| Theme | Sub-Themes |
|---|---|
| Theme 1: Understanding of UDL | • Resonance of existing inclusive practices with UDL<br>• UDL as a framework, philosophy, and tool for inclusion<br>• Systemic change for UDL implementation |
| Theme 2: UDL supporting inclusive education | • Inclusive education and UDL cannot exist independently of each other<br>• UDL addressing barriers to learning and facilitating inclusion<br>• From mere access to participation and inclusion |
| Theme 3: UDL in teacher-training initiatives and capacity building | • Conceptual shifts<br>• Capacity building of government officials<br>• Teacher education methods<br>• Importance of context<br>• Use of technology and online resources |

### 3.1.1. Resonance of Existing Inclusive Practices with UDL

The participants related how it was relatively easy for them to embrace and implement UDL as they could already relate to its principles from having applied inclusive practices to address diversity in their different contexts. They found the '*traditional marginalisation of, especially, disabled students*' (P2) to be unacceptable. They then looked for ways to address this exclusion. As they sought greater equity and access to education, they began to use various differentiation strategies, which they only realized afterward were similar to the UDL principles, although they did not identify their practices as UDL at the time. As one participant recounted:

> '*Having been a teacher who has taught in the classroom for a very long time, we never articulated these as UDL principles. . . UDL practice has been there for some years, but to call them as UDL is something I have heard over the past two years.*' (P1).

The recognition was so strong for another participant who narrated how UDL principles embody what she had always tried to achieve with her students, saying:

> '*I thought this [UDL principles] could have been written by me!*' (P10).

Another participant also expressed her passion for the inclusion of all students as something she promoted and worked hard to achieve throughout her entire teaching career, although she only learned about UDL much later through her reading and self-teaching:

> '*So that is me. That is what I have been trying to do the whole all of my life; that at least learners have access to education because education is the key that unlocks all doors. It is the pivotal right and all other rights become possible if this right is granted. . . UDL is a way of thinking, a philosophy to me about teaching and learning that gives all learners access.*' (P2).

### 3.1.2. UDL as a Framework, Philosophy, and Tool for Inclusion

The participants viewed UDL as both a philosophy that underpinned inclusive teaching approaches and a tool to operationalize the goals of inclusive education. They described it as both a "theory" and a "framework" to achieve inclusive education. As attempts were made by way of inclusive education to get children with disabilities into the mainstream, there was recognition that much broader exclusionary forces were at stake, and UDL had the potential to address these issues and facilitate inclusion.

UDL helped them to describe their own differentiation practices and gave them a framework that they could rely on to develop their teaching practices further. UDL provided a holistic framework to plan, implement, monitor, and understand teaching practices that they had already found worked well for diverse learners. For example, when some of the participants came across materials and guidance from CAST, they highlighted the capacity of a UDL approach to excite and inspire; as one of them pointed out:

*'. . . this just felt like such a useful framework. It was just filling me with. . . inspiration to take it into my new work as a teacher.'* (P5).

Participants noted that, given that the concept and practice are not unfamiliar, care needed to be taken that UDL does not become a meaningless buzzword where there is *'a whole lot of confusion about this new terminology'* (P1) and argued that the introduction of new concepts should be linked and grounded in what educators already know and then expanded from there. One participant noted that it is the practice of and not the terminology of UDL that matters:

*'For me, whether you call it UDL or not, it doesn't matter. It's about unpacking some of those concepts: How do you engage? How do you facilitate action? How do you present your material in a wide variety of ways? . . . that you really respond to the diversity of the classroom.'* (P1).

In India, there was a concerted effort to relate UDL to existing practices such as backward design and responsive teaching by a participant who described the approach as:

*'a kind of a crosswalk between UDL, backward design and responsive teaching. . . to look at. . . these three (principles) and we use this then to design lesson plans.'* (P3).

In addition, UDL helped educators move from a medical model understanding of disability to a human rights, diversity, and inclusion perspective. Teachers were supported *'to break away from some of the ways that they've been socialised into thinking about the location of the problem, or seeing disability as something that needs to be fixed, and just seeing disability as part of natural human variation'* (P9).

Importantly, the idea that UDL is only for children with disabilities was dispelled, with emphasis rather on its benefit for all learners. Participants brought to light other intersections that require attention when talking about inclusion and UDL, such as the need to develop *'culturally sustaining pedagogy or anti-racist teaching or restorative justice'* (P9). UDL could also empower and emancipate previously disadvantaged groups. For instance, one participant noted that *'in Canada, UDL is used to support inclusion of indigenous populations and working on the reservations (to modern education) that they used to have'* (P11). LMICs could also use such a strategy to include those left out of mainstream education.

### 3.1.3. Systemic Change for UDL Implementation

The participants believed that the implementation of UDL requires a shift in thinking and practices across all levels of the education system, from the policy level down to the classroom. Reflecting on the challenges of changing the system, one participant likened the introduction of UDL to *'trying to turn a ship that has been moving in a particular direction for a few hundred years'* (P3). They argued that:

*'it is not enough and it will not be effective to implement changes at policy level only. Rather, these changes need to percolate right to the grassroots level, into every single classroom.'* (P5).

Furthermore, participants observed that there needs to be a shift in the way disability is viewed and understood within the larger education ecosystem:

*'There is also need to relook at what it will take to shift the conceptual understanding from a medical model to a social or human rights model, and that needs to be looked into the context of the larger education system.'* (P1).

Concerns were raised by some of the participants about governments making efforts to make education in their country more inclusive but lacking a clear understanding of how that can be achieved. They called for a shift in understanding at the government level. For example, in India, the government is reported to be aligning education policies with the UNCRPD and the SDGs. This creates an opportune time for UDL in India. However, participants cautioned that, although the policy framework is improving and funding is being made available, an understanding of UDL is largely lacking. The limited

understanding of UDL both at the country level and by educators was a common concern among participants. One participant remarked:

*'Everyone is talking about it but unaware of what is needed to support it. The heaviness of what is needed is not understood yet.'* (P10).

*3.2. UDL Supporting Inclusive Education*

The participants recognized the close link between UDL and the implementation of inclusive education. There was a link made between the way in which UDL creates pathways for learning to enable participation and success for a much wider range of diverse learners.

3.2.1. Inclusive Education and UDL Cannot Exist Independently of Each Other

The participants saw UDL as a facilitator for inclusive education and agreed that inclusive education cannot be complete without UDL. They made statements like *'it is a question of chicken and egg and very much related'* (P12) and *'UDL is like the other side of the coin'* (P2). UDL was seen as a means to ensure education for all. Another view was that inclusive education is a broader umbrella concept and UDL is an instructional approach that provides an operational framework to implement flexible teaching in inclusive education:

*'... to see how UDL can be actually seen as almost like an umbrella and that it's a way to kind of organize some of the other (inclusive education) initiatives that are taking place at your school.'* (P9).

3.2.2. UDL as Addressing Barriers to Learning and Facilitating Inclusion

The potential for UDL to enable teachers to recognize the barriers within the design of the lesson or the learning environment was emphasized by calling UDL a 'powerful lever':

*'I think that UDL is a really potentially powerful lever in designing inclusive learning environments. I think that the framework really prompts teachers to think about what are the barriers and the design of learning experiences.'* (P9).

According to the participants, UDL enables teachers to start with the learning goals without focusing on the perceived limitations of the learners, *'making sure that all students are feeling welcomed'* (P9). Such an approach speaks to the potential of UDL to address stigma and low expectations of learners with disabilities, as highlighted by one participant:

*'the core value proposition of UDL, which is to flip that and say no, it's the curriculum that is broken, that is inflexible, and just that inherent kind of belief system of valuing everyone's unique genius...'* (P8).

3.2.3. From Mere Access to Participation and Inclusion

The implementation of UDL, linked to quality inclusive education that addresses participation and achievement, was a priority for participants. *Having UDL within the education system is the one thing that can make a difference between participation and marginalization so you're either on one extreme or the other* (P12). They shared that UDL impacts the curriculum and learning materials by foregrounding flexibility of representation, engagement, and expression, thereby making learning environments more inclusive. At a broader level, inclusive education was equated to:

*'... inclusion to life; preparation for inclusion to all areas of life, and UDL brings together the aspects that the different approaches to inclusive education (in the classroom) have been trying to address separately over the years.'* (P10).

Through the holistic instructional approach, all learners can feel included and enjoy learning, and the teachers can *'make learning not like a prison, but like a holiday resort'* (P2). Another participant shared the same sentiments on UDL enabling access to the learning environment: *The whole teaching process needs to be accessible. We need to look at learners and see how then the learner should be able to access a learning environment, so simply it is a*

*curriculum differentiation* (P2). UDL promoted participation for all learners in the classroom: *[It's] not just tokenistic, that these learners are in school and it's just about statistics, but actually they meaningfully participate in the education system because it's designed in such a way that is responsive to their specific needs* (P12). This indicated that UDL shifts the focus from the number of children enrolled to their meaningful participation.

*3.3. UDL in Teacher-Training Initiatives and Capacity Building*

There was unanimous agreement amongst the participants that capacity building of teachers, education officials, and policymakers is a critical element of the implementation of UDL. They acknowledged that important conceptual shifts needed to take place. This would require capacity building not only of teachers but also of education officials and policymakers to support this work. Teacher education should consider the specific context, and this includes being aware of the available technology and resources.

3.3.1. Conceptual Shifts

According to the participants, the shift begins with understanding UDL: *'The teachers need to understand what UDL is. What's the essence behind it? What are the principles behind it?* (P1). This requires shifting understanding that inclusive education is not only relevant to students with disabilities but to all: *'That is the challenge, to deliver the message that inclusive education is about good education for all children, and it doesn't matter if they have a disability, whether it's girls, boys, etc.'* (P10). The same shift in understanding UDL was echoed by another participant who emphasized that: *'. . . for UDL you're looking even beyond just children with disabilities to looking at all levels of marginalization.'* (P11).

Importantly, in some LMIC contexts, participants felt that teachers need to shift their teaching from a transmission, textbook-based mode to one where they encourage their learners *'to problem solve and to apply their knowledge, rather than just recall'* (P3). It was pointed out that the levels of, and requirements for, teacher education are very diverse in different contexts. Training in many LMICs is limited and may only impart basic knowledge without an understanding of diversity. For instance, one participant advocated for using UDL in teacher training:

> *'When teachers don't have that pedagogical base it's hard to build on top of that with UDL, for example, so I think the overall sort of nature of teacher training program is potentially a challenge. You know, we ideally want to plug UDL into pre-service training, but if pre-service training isn't widely required, then what are we plugging it into?'* (P11).

It was also pointed out that the way in which educators understand disability must be addressed since: *'teachers are part of the society, and we know attitudes, cultures and beliefs have an impact on how teachers respond to diversity in the classroom'* (P1). Using a social model approach, the social determinants influencing inclusive education and the participation of children with disabilities in schools can be addressed. *'They've been socialized into thinking about the location of the problem or seeing disability as something that needs to be fixed. . . (instead of) just seeing disability as part of natural human variation you know, and just kind of reframing that'* (P9). A shift needs to happen in the minds of teachers from *'thinking of training for inclusion of children with disabilities. . . (to) training for inclusion for teachers to deliver good education'* (P1).

3.3.2. Capacity Building of Government Officials

Given the scarcity of UDL adopters and a lack of understanding of the concept, it was suggested that capacity building has to start at the ministry level for it to have an impact on the countries' inclusive education policies. The buy-in of policymakers was also emphasized as necessary for schools and teachers to get guidance and support and for clearance to spend resources on capacity building without fear of *'getting in trouble for spending their public dollars in this way'* (P9). One participant noted that innovative resource mobilization is required to support online training, particularly where trainees might be unable to afford specific resources such as textbooks. Plans also need to be made

to remunerate external trainers. Hence, the involvement of government authorities and policymakers is also crucial for financing and sustaining UDL training programs.

Government officials tasked with teacher training need to be made aware that UDL should be included in the policy for designing pre-service teacher training and not be added as an afterthought that requires costly adaptations to the system: *'The push for design in the early stages within the education system is what seems to be desirable.'* (P12). One participant suggested that training needs to be approached in a *'multi-sectoral interdisciplinary way. . . because it is not only about providing screening, but providing contextual information to teachers about what are the resources and support within the system which is available to them'* (P1).

### 3.3.3. Teacher Education Methods

Participants expressed a need to move from the common 'theoretical' UDL training happening in LMICs to more practical, hands-on approaches:

*'We have thousands of experts on education who speak on UDL, but always on a very theoretical note, and it's very difficult for trainees afterwards to apply.'* (P4).

The practical training has to be bolstered by evidence of 'true stories' of successful implementation of UDL:

*'I think for a lot of teachers who might be feeling like 'Oh, UDL, is it for me?', I really strongly believe that seeing that evidence, showing what works for students, showing examples of work and also video footage of classrooms, to really be sharing what's possible, I think that can be a really powerful hook for teachers.'* (P9).

Participants also noted that training needs to be a process that is accompanied by a plan and ongoing support:

*'I think that any time you do a training it's important for the individuals who are bringing you in to think about, okay, what are we going to do next? After this training, how are we going to continue to support those ideas so that they will grow and so that people have an opportunity to, you know, ask questions when things come up?'* (P7).

Training manuals or *'toolkits'* that are *'simple and straight to the point'* (P2) were identified as an effective approach that would provide teachers with materials to work with at pre-service and in-service levels. Collaboration among the adopters of UDL in LMICs through ongoing mentoring and peer support was emphasized, with suggestions for building *'professional learning networks'* (P9) using means that are viable in each specific context, such as WhatsApp groups. Peer learning and the sharing of information and UDL practices can also be encouraged through *'instructional rounds'* (P9), where teachers observe each other's lessons while implementing UDL. This was emphasized by one participant:

*'When they hear it from their own colleagues, and hear that this is working in my particular context, in my particular school, that's oftentimes what gets teachers to think like, okay, maybe I want to do it.'* (P8).

Collecting evidence with teachers on the practicality of UDL also helps to motivate them, as observed by one participant in a context where they worked with teachers: *'Seeing student work that really exceeded teachers' initial expectations was a huge motivator for them in terms of learning more about UDL and wanting to continue to experiment with UDL in their classroom.'* (P8).

While participants shared information about some promising training courses on inclusive education that are incorporating UDL in their content, they noted that as part of entrenching UDL in teacher education, there is a need to include a full module on UDL in pre-service training and to ensure ongoing support for in-service teachers. *'Teacher training needs to move away from a 'project-based approach' where teachers receive short workshops or seminars that are quite 'theoretical. . . or may not be tied closely to the local context'* (P11).

The introduction of UDL in LMICs is frequently initiated through international collaboration. One participant learned about the approach on an exchange visit to Canada and

then implemented it in LMIC contexts. There was general recognition that such collaboration needs to be carefully managed so that *'on- the-ground organisations do the actual UDL training and carrying the work forward'* (P10). There is a need to build *'capacity so that trainers who live in these LMICs are themselves equipped to be their local UDL experts, and be able to really support UDL practice in their own authentic context'* (P8).

### 3.3.4. Importance of context

Participants emphasized context-relevant and context-specific teacher training, starting from where teachers are, as *'not everybody comes in at the same level and not everybody has the same needs as to what they get out of the course'* (P7).

They suggested addressing this challenge by involving teachers in developing the UDL training material to meet their context-specific needs by asking questions like *'Where can you start?'* and *'How do you think UDL will make a difference?'* (P10). Participants involved in teacher training emphasized:

> *'. . . learning from local contacts about what training methodologies are going to be most helpful, and if there are pieces in our training. . . that. . . will need to adjust to make it more culturally competent for a particular setting and for a particular country's concerns.'* (P7).

The capacity building of teachers also needs to be supported by families and communities *'in terms of drawing the support and really making parents as partners in the educational planning of children in the classroom'* (P1). The fact that there is a community they can draw upon *'is something that needs to be well understood and then proposed as part of the training. . . delivered to teachers'* (P1).

That way, an ecosystem, which will take different forms depending on the context, can be leveraged. *'In every culture and context, the situation will be different and we need to take note of that'* (P10).

### 3.3.5. Use of Technology and Online Resources

Participants noted that given that *'the future is blended learning'* (P5), there is a need to consider the best way to include teachers in online learning. Many free resources are available online, and those from CAST have been found particularly useful. However, teachers might need to be motivated to use technology if they lack confidence in the approach.

According to the participants, the issue of digital literacy in UDL relates to both the teachers who are adopting the method as well as to their learners. As regards teachers, they may experience limitations in accessing UDL training that is offered on digital platforms: *'Many, probably 70 to 80% of the teachers in low- and middle-income countries may not necessarily have the skills with digital and online learning.'* (P12). It was also felt that training in digital literacy should start at the pre-service stage in order for teachers to better appreciate inclusive practices. Moreover, the materials presented during training might need to be adapted for computer and mobile-device users, depending on the technology trainees have access to. Importantly, internet connectivity needs to be considered during training as some teachers might be left behind while others benefit.

> *'[What] I did find out in my work around the world is that not everybody has good access to online instruction, so sometimes we need an in-hand on paper tool. Yeah, always keep that in mind.'* (P7).

As part of capacity building, teachers need to understand that UDL is not all about technology, although technology might ease implementation in some instances. *'Decoupling it [UDL] from technology is so important'* (P9). Teacher training can be enhanced through blended learning, which follows some models that have worked in high-income countries, such as the CAST programs, which use *'online modules that allow for self-paced learning. . . various kinds of coaching structures and hands—on learning opportunities—it's a good mixture of elements.'* (P8).

Participants cited mobile phones, WhatsApp, Google Apps, Microsoft Teams, videos, laptops, and local materials as some of the most viable technologies that could be used in LMICs. Low-tech resources can be recycled from warehouses, businesses, and other organizations, which underlines the central role of the community in sourcing materials.

*'You could use community rehabilitation workers and connect them to businesses to get materials, and perhaps this could be done through a non-profit. And then think how to get the materials out there [to schools].'* (P7).

The need for technology to address the needs of all learners, including those with disabilities, was emphasized. For instance, *'there are excellent apps that have been developed in India to help children who are non-verbal'* (P5). The need to contextualize the use of technology was highlighted, as well as the view that no one technology fits all contexts. The example given was that *'we've seen many, many situations where tablets are introduced into classrooms, but the children don't have electricity at home, so that tablet stays in the classroom behind lock'* (P11). Improving access to technology needed to be matched with the empowerment of people and communities in digital literacies and upskilling teachers; *'otherwise you're just ensuring better access to Netflix!'* (P8).

Training materials need to be repackaged for online delivery that will suit a diversity of audiences as well as to download when data is accessible for later use offline. Mobile technology could be exploited for this purpose, given its prevalence in LMICs. There is ample evidence of the power and reach of online resources:

*'I have just used the internet. I have self- taught myself and read a lot about UDL. I have just learned UDL, but I do not have a qualification in that.'* (P2).

While lack of resources was a real problem, participants felt that it should not be used as an excuse not to begin the processes of change:

*'That is the excuse that we always use in Africa, that we don't have resources and I always say, 'Come on, the first resource is you.' All other resources will fall into place. But the first thing is the political will.'* (P2).

What was also evident was the enormous influence of CAST and their online materials in spreading the philosophy and practical implementation of UDL. Most participants read CAST materials not only for themselves but also to include in the training programs they offered. They *'were able to look at CAST, website, etc. and use that extensively'* (P5). Based on this experience, teacher educators have begun to include UDL in their formal curriculum.

*'I think, also incremental ways of actually achieving that so definitely providing the capacity building training through online models is a good way to go, but we need to recognize the gaps and see how we make this incremental other than just outright because we may miss out quite a few.'* (P12).

Given that over-technologizing UDL is unrealistic in low-resource settings, the participants recommended focusing on low-tech and no-tech approaches. Participants pointed out that many low-tech resources exist in remote areas and that ways to integrate these technologies with information and communication technology are needed.

## 4. Discussion

It becomes apparent that inclusive education and UDL cannot exist independently of each other, where UDL is seen as an instructional approach that provides an operational framework to implement flexible teaching in inclusive education. However, UDL goes further than just a framework; it is also a theory or philosophy to inform inclusive education. As such, it is not an unfamiliar approach in many LMICs, even though the name UDL may not be familiar. Thus, while the introduction of UDL in LMICs is frequently initiated through international collaboration, attempts to enhance the adoption of UDL in these contexts should recognize the prior knowledge existing in these contexts rather than consider them virgin territory.

The recognition of UDL-related differentiation practice is important to avoid the risk of UDL becoming a fancy terminology with little practical implication, or what Karisa [15] called "another slogan on the street of inclusive education". One way of navigating this risk is for the proponents of UDL to build on the familiar, existing efforts already happening in LMIC that use UDL without necessarily calling them so. It was apparent from the participants' previous experiences of practicing as teachers and of working with teachers in their current roles as leaders and consultants that most teachers in LMIC understand and have insights about ways to include students with diverse learning needs in their classrooms. Although they may not be aware of frameworks like UDL, there are some efforts and practices in place already incorporating some UDL principles. Previous studies from LMIC reported similar findings [16,17].

The existing inclusive practices can be leveraged and/or further strengthened by UDL. UDL can be implemented as a framework that provides guidelines and a language of reference for capacity building of teachers and other stakeholders while not undermining existing inclusive practices and assets in each context. This is especially crucial in LMICs, where most of the existing systems are based on Western ideologies, which may not be compatible with local practices [17,18]. Although some scholars have contested the potential of UDL to improve access and participation in learning for students in LMIC due to the hegemonic, institutionalized ideologies of the education systems that promote capitalism and neoliberalism [15], we see a positive side of UDL in these contexts. That is, if UDL is implemented in a context-sensitive way, as advocated by the study participants, the teachers and education administrators can promote ways of engagement, representation, and action and expression that are based on local and indigenous knowledge systems [19] while making use of locally available teaching materials that learners can relate to. Although technology can be important, teachers need support to comprehend that UDL encompasses more than just sophisticated technology. Low-tech alternatives can be developed for use, as exemplified by Song [16] and Braun et al. [19] in South Africa and Tanzania, respectively.

The current absence of a systematic approach to teacher training in UDL in LMICs is a pressing concern. Despite some expressed interest of governments in UDL, it has yet to become a prominent educational priority. Therefore, the involvement of government authorities and policymakers is pivotal in providing financial support and ensuring the sustainability of UDL training programs. As the concept of UDL is currently likely to be implemented in an ad hoc manner in LMIC, bespoke training in UDL would provide teachers and other education stakeholders with a solid framework on which to base their practices. It would be beneficial if among the trainers were UDL practitioners from LMIC who understand the contextual dynamics facing education in LMIC, including the shortage of resources and overcrowded classrooms [20,21]. Furthermore, teacher training can be enhanced through blended learning. The training materials should be repackaged for efficient delivery, ensuring that they are downloadable for offline use, especially in the case of online training. Strengthening practical training through, for example, the demonstration of successful UDL implementations is imperative. It is also important for UDL to be integrated into policy for the development of pre-service teacher training rather than being treated as an afterthought that necessitates costly adaptations to the existing system.

It is evident that UDL calls for conceptual shifts in how education stakeholders, including teachers and education leaders, view learning. As noted by Rao et al. [22], UDL "has the potential to prompt practitioners to uncover and wrestle with their preconceived attitudes and beliefs". This needs to happen across the different levels of the education ecosystem, from government ministries down to the classroom. The areas requiring this change range from a shift in the understanding of student diversity, issues of marginalization, and attitudes towards students with different learning needs, especially those with disabilities, to an understanding of UDL and its benefits and how to implement it in a particular context. Training for these education stakeholders should facilitate a conceptual shift from the approach where the learner is expected to adapt to the education system to the UDL approach, which emphasizes the need for the system to adapt to the learner. This

is a similar argument to that advanced by Karisa [15], suggesting the need to change the education system holistically to enhance the educational inclusion of students with various learning needs. Although UDL may be considered a leading concept towards inclusive education, it is one method among many in the toolbox of inclusive education. This echoes the sentiment of the International Disability Alliance (2021) [23] that UDL does not replace the need for reasonable accommodation, support services, or assistive devices.

## 5. Conclusions

The relevance and fit of UDL to LMIC contexts depend on these contexts, making the concept "their own" and tailoring it to their specific needs. LMICs have unique needs that may not resonate with those of high-income countries. Therefore, when introducing UDL to LMICs, it is crucial to approach the concept critically and be accountable to the contextual realities rather than pursuing a Western ideal of its implementation. It should be noted that the concept of UDL is not new in LMICs, although the name may be. Consequently, the encounter between the proponents of UDL from high-income countries and education stakeholders in LMICs should be mutually enriching rather than being top-down, where those from high-income countries 'impose' the approach on those in LMICs. While acknowledging the limitations of our small sample size and the use of purposive sampling in this study, potentially favoring a specific perspective on UDL, we remain confident that the study has brought forth valuable insights for the implementation of UDL in LMIC. Given the current global momentum towards inclusive education, wherein UDL serves as a recommended strategy for implementing this initiative [4], it would be beneficial for future research to shift its focus to the teachers themselves to analyze their experiences as they implement UDL in LMICs. This is particularly salient because what researchers describe as UDL may differ from what happens in the classroom [24]. Since the participants in the current study consisted of UDL experts from leadership and consultancy roles, as well as researchers and representatives of international agencies and NGOs, exploring the teachers' experiences would provide valuable insights into the practical challenges and successes associated with UDL implementation at the grassroots level in LMICs.

**Author Contributions:** Conceptualization, J.M., A.K. and C.K.; formal analysis and writing—original draft preparation, J.M., A.K. and C.K.; writing—review and editing, J.M., A.K. and C.K. All authors have read and agreed to the published version of the manuscript.

**Funding:** The research that underpins this paper was funded through the CBM Community Based Inclusive Development (CBID) Initiative's Global Inclusive Education section, Germany.

**Institutional Review Board Statement:** The study was conducted according to the guidelines of the Declaration of Helsinki and approved by the Ethics Committee of the University of Cape Town (HREC REF: 148/2021). The date of approval was 11 March 2021.

**Informed Consent Statement:** Informed consent was obtained from all subjects involved in the study. Written informed consent to publish this paper has been obtained from the subjects.

**Data Availability Statement:** The data were used under license for the current study; restrictions apply to their availability, so they are not publicly available.

**Acknowledgments:** We are grateful to Sian Tesni, Global Inclusive Education Advisor, CBM, for her guidance, support, and input in this work.

**Conflicts of Interest:** The authors declare no conflict of interest.

## Appendix A

*Interview Schedule*

1.  What is your experience of and understanding of UDL? Have you had any specific training on the topic? Have you offered any training through your organisation?
2.  How do you see UDL as supporting inclusive education? How do they connect in policy, and implementation?

3. What are your views on current inclusive education training initiatives in LMIC? How can UDL be incorporated into teacher education?
4. What do you think are the training and capacity building needs for UDL in LMIC?
5. What recommendations would you make for online learning for UDL in LMIC? How do you envisage that these could be resourced?
6. What technology do you think can be used in UDL in LMIC?
7. In the classroom?
8. In what ways do you think that UDL can address equity issues and address discrimination?
9. What do you see as the challenges and potential of UDL in LMIC?

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
