# Peer review of "Implementation of Universal Design for Learning in Low- and Middle-Income Countries: ‘I Thought These Principles Could Have Been Written by Me’"

_disabilities, doi:10.3390/disabilities3040043_

Round 1
Reviewer 1 Report
Comments and Suggestions for Authors
This is a very pertinent paper that fills an important gap in the literature. It will make an important contribution to the field. I recommend it be published but requires major editing, in the sense that the methodology section is weak at present. The outcomes are strong, bold, and have immediate pertinence to the field, but the data collection and analysis are incredibly weak, and this has the danger of making this piece appear as an opinion piece, a very sound one - but an opinion piece, nonetheless. It can be salvaged and become very strong. There seems to be nothing inherent faulty with your data, but you must strengthen your data analysis section. I would begin by adding a strong section on theoretical stance and assumptions. I would then rewrite the data analysis section entirely. Please state the type of methodology/ methodological tradition used, and the tools chosen. Secondly, coding is not simply a matter of using Word track changes. An extensive edit/ rewrite is required for the section. It must be informed by literature and stand the test of qualitative scientific rigour. At present it is just too methodologically weak to be published. I encourage you to invest time in doing so as the piece otherwise has the potential to be very pertinent and noteworthy,
Comments on the Quality of English Language
Few issues with the expression or flow of the text. A minor copy edit is recommended.
Reviewer 2 Report
Comments and Suggestions for Authors
Dear authors,
Thank you for your interesting paper. I enjoy reading it.
I think it is a good, relevant, organized, well-structured and convincing paper. I made small suggestions that I think can improve your article. Please see the document attached.
Best
